# HEDP: A Hybrid Energy-Distance Prompt-based Framework for Domain Incremental Learning

Yu Feng [1]  Zhen Tian [2]  Haoran Luo [3]  Xie Yu [4]  Diancheng Cheng [1]  Haoyue Zheng [1]  Shuai Lyu [2]  Ping Zong [2]  Lianyuan Li [1]  Xin Ge [1]  Yifan Zhu [2]

## Abstract

Domain Incremental Learning is a critical scenario that requires models to continuously adapt to new data domains without retraining. However, domain shifts often cause severe performance degradation. To address this, we propose Hybrid Energy-Distance Prompt, a domain-incremental framework inspired by Helmholtz free energy. HEDP introduces an energy regularization loss to enhance the separability of domain representations and a hybrid energy-distance weighted mechanism that fuses energy-based and distance-based cues to improve domain selection and generalization. Experiments on multiple benchmarks, including CORe50, show that HEDP achieves superior performance on unseen domains with a 2.57% accuracy gain, effectively mitigating catastrophic forgetting and enhancing open-world adaptability. Our code is available here.

## 1. Introduction

To ensure the long-term viability of deep learning models in mission-critical systems, they must remain robust across diverse environments. For example, autonomous-driving (Wu et al., 2023; Tahir et al., 2024) object detectors trained on clear-weather data fail under adverse conditions. Domain-incremental learning (DIL) (Shi & Wang, 2024; Wang et al., 2023a; van de Ven et al., 2022) addresses this challenge by treating datasets from different distributions as distinct domains and training the model sequentially across them. When scaling up to large-parameter models(Song et al., 2025), DIL can substantially reduce training costs(Zhou et al., 2025) while continually enhancing the model's gener-

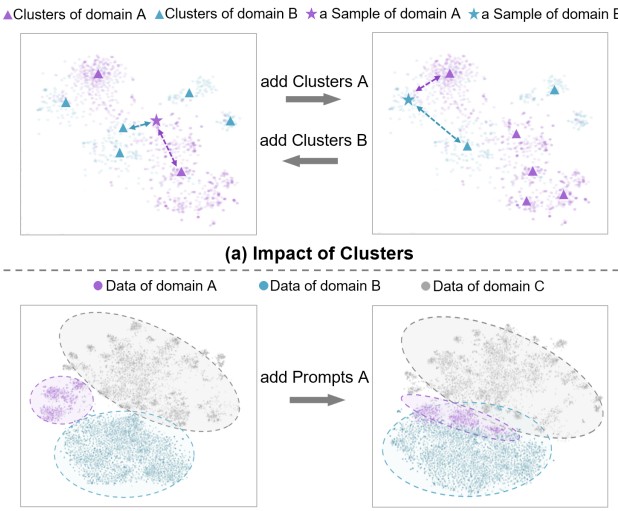

*Figure 1.* t-SNE visualization of the feature space illustrating knowledge reuse challenges: (a) Impact of Clusters: simply adding cluster centers (triangles) creates ambiguous boundaries where samples from Domain B (cyan) are spatially closer to Domain A's clusters. (b) Impact of Prompts: optimizing single-domain prompts independently pushes distributions apart, creating overlap in the projected space that confuses the decision boundary.

alization ability.

In DIL, previously seen domains are known (van de Ven et al., 2024; Mannering & Jones, 2021), while unseen ones are unknown (Zhou et al., 2024; Nicolas et al., 2024), and the goal is to maintain consistent inference across both. Forgetting stems from the non-stationary nature of DIL, where sequentially learned, distributionally distinct domains overwrite prior knowledge—e.g., indoor-trained models degrade on outdoor data (Lomonaco & Maltoni, 2017; Pellegrini et al., 2021).

While CP-Prompt (Feng et al., 2024) mitigates forgetting and Mop-CLIP (Nicolas et al., 2024) improves generalization, balancing inference across known and unknown domains remains challenging. Feature visualization (Fig. 1) shows that overlapping domain spaces make distance-based selection unreliable, and intra-domain prompts risk overfitting, increasing inter-domain confusion. This highlights

[1]China Mobile Research Institute, China [2]Beijing University of Posts and Telecommunications, China [3]Nanyang Technological University, Singapore [4]Beihang University, China. Correspondence to: Yifan Zhu <yifan_zhu@bupt.edu.cn>.

*Proceedings of the $43^{rd}$ International Conference on Machine Learning*, Seoul, South Korea. PMLR 306, 2026. Copyright 2026 by the author(s).

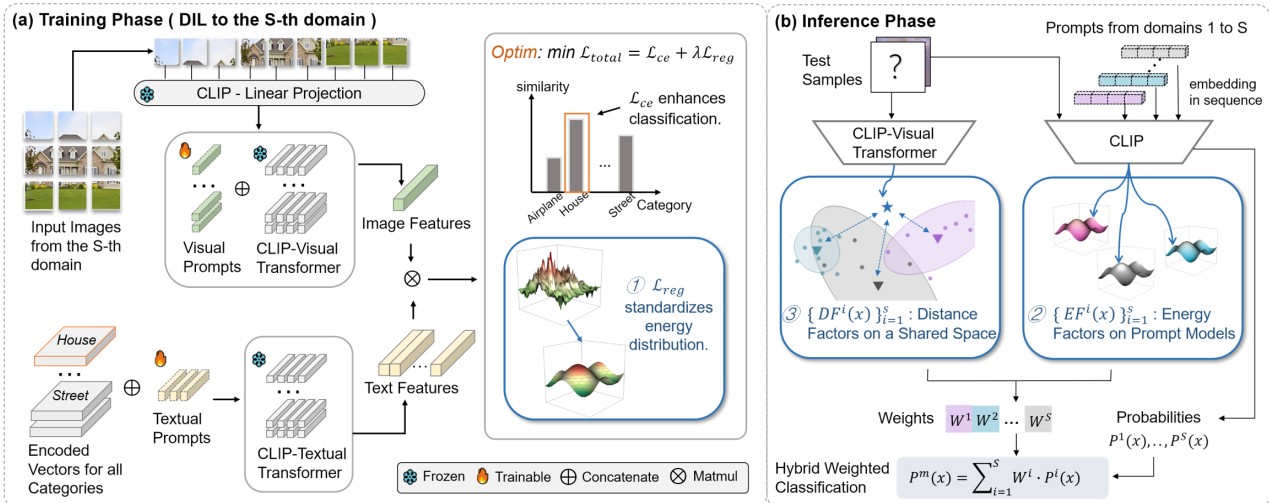

*Figure 2.* The overall architecture of HEDP. (a) Training phase: In DIL, single-domain visual-textual prompt parameters are optimized using data solely from the current domain, with a frozen CLIP model, aiming to minimize $\mathcal{L}_{ce}$ and $\mathcal{L}_{reg}$. (b) Inference phase: The model classifies samples by applying a hybrid weighted strategy to the prompt models, which utilizes energy factors $EF(x)$ and distance factors $DF(x)$ to determine weights. The main designed components are represented by ①, ②, and ③.

two key challenges: mitigating inter-domain overlap during training and selecting domain knowledge at inference to maximize generalization.

To address these challenges, we seek to reinterpret inter-domain differences through a physical lens. A domain's data characteristics are determined by their statistical distribution in feature space, just as an energy field in physics describes how forces vary in magnitude and uniformity across locations. The structure of a data distribution closely mirrors such an energy field. Inspired by Helmholtz free energy(LeCun et al., 2006; Liu et al., 2020), we model the data distribution as an energy landscape to delineate boundaries between domains.

We propose the HEDP architecture with two key components: Energy Regularization Loss and the Hybrid Energy-Distance Weighted mechanism. **The first** improves discriminability between in- and out-of-domain knowledge, addressing overlapping or disordered energy curves that hinder domain attribution and knowledge reuse. **The second** stabilizes domain selection by combining energy and distance, mitigating overfitting or prompt bias and improving performance near overlapping domain boundaries. The main contributions are:

- An energy regularization loss that stabilizes prompt-induced energy distributions and reduces inter-domain ambiguity;

- A physics-inspired hybrid inference mechanism that surpasses traditional prompt ensembling by combining prompt energy and shared-space distance to improve knowledge retention and generalization.

- Comprehensive validation of HEDP on three benchmark datasets, highlighting its effectiveness in improving inference across unknown domains.

**Conflict of Interest Disclosure.** The authors declare no financial conflicts of interest related to this work, including any evaluation of models, products, or services from financially related organizations.

## 2. Related Work

### 2.1. Domain Incremental Learning

Domain Incremental Learning adapts to shifting distributions (Huang et al., 2024; Rui et al., 2023). Prompt-based methods improve efficiency by updating prompts instead of model weights (Feng et al., 2024; Nicolas et al., 2024; Wang et al., 2023b; 2022b;a; Douillard et al., 2022), reducing cost and training time (Ke et al., 2023; Zhou et al., 2022). A key challenge is balancing personalization and generalization. While prompt aggregation strategies (Xu et al., 2024)have been explored to enhance robustness against unseen domains, existing methods still struggle to reconcile this with domain-specific accuracy. Methods such as CP-Prompt (Feng et al., 2024) and ESN (Wang et al., 2023b) achieve strong performance in known domains but struggle to generalize to unseen domainss, while MoP-CLIP (Nicolas et al., 2024) prioritizes unseen-domain inference at the expense of known-domain accuracy due to coarse clustering.

### 2.2. Energy-Based Model

Energy-Based Models (EBMs) (LeCun et al., 2006; Liu et al., 2020) assign scalar energies to samples, enabling

OOD and open-set detection (Chen et al., 2024; Al Rahhal et al., 2022), yet remain underused in incremental learning. Existing attempts—such as ELI (Joseph et al., 2022), which models task-wise energy manifolds, and ESN (Wang et al., 2023b), which introduces a temperature-controlled metric—illustrate initial explorations of EBMs in DIL, while leaving room for more general and robust designs.

# 3. Preliminaries

## 3.1. Problem Definition

To simulate a domain incremental learning (DIL) scenario, we use an image classification task with $S$ domains, each containing dataset $\mathcal{D}^i$. The model $\mathcal{M}$ trains sequentially, starting with $\mathcal{D}^1$ and then processing $\{\mathcal{D}^2, \mathcal{D}^3, \ldots, \mathcal{D}^S\}$ one domain at a time, where $i \in \{1, \ldots, S\}$ denotes the index of the domain. During new domain training, the model cannot replay previous domain samples but can retain a small set of their prompt parameters. Even after learning $\mathcal{D}^S$, $\mathcal{M}$ must infer on known domains $\{\mathcal{D}^1, \mathcal{D}^2, \ldots, \mathcal{D}^S\}$ and generalize to unknown domains $\{\mathcal{D}^{S+1}, \mathcal{D}^{S+2}, \ldots\}$. Our inference paradigm strictly requires the model to be agnostic to the test sample's domain origin.

Specifically, the dataset $\mathcal{D}^i$ from the $i$-th domain comprises image-text pairs, denoted as $(x^{i,j}, t^{i,j})_{j=1}^{N_i}$. Here, $x^{i,j} \in \mathbb{R}^{W \times H \times C}$ represents the $j$-th image sample in $\mathcal{D}^i$, with each image characterized by dimensions $W \times H$ and $C$ channels. The number of samples in the $i$-th domain is denoted by $N_i$. The textual label $t^{i,j} \in \{a, b, \ldots, z\}^{|t|}$ corresponds to the $j$-th sample in $\mathcal{D}^i$, composed of lower-case English letters, where $|t|$ indicates the length of the label. For the quantization of classification labels, they are represented as $y^{i,j} \in \{1, \ldots, U\}$, with $U$ representing the total number of categories. Consequently, the dataset can be formulated as $\mathcal{D}^i = \{(x^{i,j}, y^{i,j})\}_{j=1}^{N_i}$.

To enable continual adaptation, we adopt domain-specific prompting. Let $P^* = \{P^1, P^2, \ldots, P^S\}$ denote the set of prompt parameters learned across domains. Since each domain is trained in isolation, the learning objective seeks optimal prompts that minimize the cumulative loss over all domains:

$$F_{\text{obj}} = \sum_{i=1}^{S} \min_{P^i} \mathcal{L}^i\big(\phi\big(X^i \mid M_{\text{pre}}, P^i\big), Y^i\big), \quad (1)$$

where $X^i = \{x^{i,j}\}_{j=1}^{N_i}$ and $Y^i = \{y^{i,j}\}_{j=1}^{N_i}$. The function $\phi(X^i \mid M_{\text{pre}}, P^i) \in \mathbb{R}^U$ denotes the predictions generated by the pre-trained backbone $M_{\text{pre}}$ conditioned on prompt $P^i$, and $\mathcal{L}^i$ is the intra-domain classification loss. This formulation captures the incremental optimization of domain-specific prompts under domain-isolated training constraints.

## 3.2. Helmholtz Free Energy

Inspired by statistics, where the Boltzmann distribution (Mc-Dowell, 1999) describes particle probabilities across energy states, and Helmholtz free energy (Liu et al., 2020) defines a system's energy at constant temperature and volume:

$$E = -kT \cdot \ln(Z), \quad (2)$$

where $k$ is the Boltzmann constant, $T$ is the Temperature, and $Z$ is the partition function, which includes the sum of all possible energy states' Boltzmann factors.

In machine learning, we analogize this to describe a sample's classification probability. Given a model's output similarity $H(x)$ for sample $x$, where $H(x)[y]$ is the similarity for class $y$, the classification probability $P(y|x)$ can be calculated via softmax, akin to Helmholtz free energy. This transforms into an energy formula:

$$P(y|x) = \frac{e^{H(x)[y]/kT}}{\sum_{y'=1}^{U} e^{H(x)[y']/kT}} = \frac{e^{-E(x,y)/kT}}{e^{-E(x)/kT}}, \quad (3)$$

the numerator reflects the energy variation of sample $x$ for class $y$, while the denominator represents the variation across all possible classes $\{y^i\}_{i=1}^{U}$. The Helmholtz free energy $E(x)$ for a sample is then defined as:

$$E(x) = -kT \cdot \ln \left[ \sum_{y=1}^{U} e^{H(x)[y]/kT} \right], \quad (4)$$

which is termed energy in subsequent sections.

# 4. Methodology

## 4.1. Overall Architecture

Fig. 2 presents the HEDP architecture, a prompt-based DIL framework designed to prevent catastrophic forgetting and improve generalization without data replay. During training, domain-specific prompts are independently optimized with an energy regularization loss to constrain energy distributions while preserving accuracy. After training, prompt parameters are frozen to retain knowledge. At inference, energy and distance factors are fused into hybrid weights to adjust predictions from each prompt model, enabling unified, adaptive inference across known and unknown domains.

## 4.2. Training Phase with an Energy Regularization Loss

To address Challenge 1, the training phase introduces an Energy Regularization Loss to optimize the prompt-based, domain-specific training process and mitigate distributional overlap. We adopt CLIP (Radford et al., 2021) as the pre-trained model $M_{\text{pre}}$ for its strong performance on multi-modal tasks. In HEDP's DIL implementation, each domain

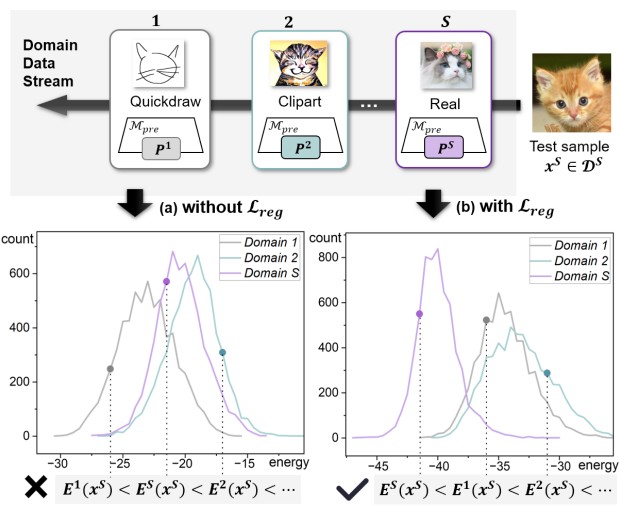

*Figure 3.* The purple curve corresponds to the Real domain, the gray curve to the Quickdraw domain, and the green curve to the Clipart domain. Sample $x^S$ from Domain $S$ is processed by prompt models across domains to obtain a set of energy values, with lower energy values indicating closer proximity. (a) Without $L_{reg}$, the sample is mistakenly ranked closest to Domain 1. (b) With $L_{reg}$, the sample is correctly ranked closest to Domain $S$.

initializes and locally optimizes its own prompt parameters using single-domain data, while keeping $M_{\text{pre}}$ frozen.

**Visual and Textual Feature Extraction.** On the visual side, a 2D image $x \in \mathcal{D}^S$ is flattened into 1D patch tokens via linear projection with positional encoding, then fed into a 12-layer transformer encoder along with visual prompts $P_v^S$. The output image features $Z_v^S \in \mathbb{R}^D$ are obtained via prefix-one-tuning (Feng et al., 2024). On the textual side, the prompt parameters $P_t^S$ are duplicated $U$ times and concatenated at the prefix of the encoded vectors for all categories, $E = \{e^i\}_{i=1}^U \in \mathbb{R}^{U \times C \times D}$, forming a new set of vectors $e = [U \times P_t^S; E] \in \mathbb{R}^{U \times (C + L_{PT}) \times D}$, where $U$ is the total number of categories, $C$ is the length of sentences for each category, and $L_{PT}$ is the length of the textual prompts. Subsequently, $e$ is input into the transformer encoder to obtain the set of text features $Z_t^S \in \mathbb{R}^{U \times D}$ for all categories.

**Objective Function and Energy Regularization.** Based on the image features and the set of text features, the cosine similarity $H^S(x) \in \mathbb{R}^U$ is constructed, as follows:

$$H^S(x) = \cos(Z_v^S, Z_t^S) = \frac{Z_v^S \cdot Z_t^S}{\|Z_v^S\| \cdot \|Z_t^S\|}, \qquad (5)$$

To optimize the current domain's prompt parameters $P^S = \{P_v^S, P_t^S\}$, we load domain-specific data in batches. Given a batch $\mathcal{D}_t$, we compute the total loss $\mathcal{L}_{total}$, composed of two terms balanced by $\lambda$:

$$\min_{P^S} \mathcal{L}_{total} = \mathcal{L}_{ce} + \lambda \mathcal{L}_{reg}, \qquad (6)$$

$\mathcal{L}_{ce}$ is a cross-entropy loss for classification, as follows:

$$\mathcal{L}_{ce} = -\frac{1}{|\mathcal{D}_t|} \cdot \sum_{(x,y) \sim \mathcal{D}_t} \ln \left[ \frac{e^{H^S(x)[y]/kT}}{\sum_{y'=1}^U e^{H^S(x)[y']/kT}} \right]. \quad (7)$$

Energy regularization loss $\mathcal{L}_{reg}$ is composed of a boundary loss $\mathcal{L}_{border}$ and a midline loss $\mathcal{L}_{midline}$:

$$\mathcal{L}_{reg} = \mathcal{L}_{border} + \mathcal{L}_{midline}, \qquad (8)$$

As shown in Figure 3, without energy regularization, the energies of $x^s$ across different prompts overlap, causing ordering inconsistencies such as $E^1(x^s) < E^s(x^s) < E^2(x^s)$ and leading to incorrect domain similarity judgments. With the regularization term $\mathcal{L}_{reg}$, the energy of $x^s$ under its true domain becomes distinctly lower, yielding

$$E^s(x^s) < E^1(x^s) < E^2(x^s) < \cdots, \qquad (9)$$

which aligns with the expectation that better domain–prompt alignment produces lower energy. Thus, energy comparison reliably reflects cross-domain similarity when regularization is applied.

**Boundary Loss: Controlling the Upper Energy Bound** The boundary loss encourages in-domain samples to lie on the low-energy side of a predefined boundary $\Theta$, thereby enlarging the separation between in-domain and out-of-domain energies. It penalizes only the portion of energy exceeding the boundary:

$$\mathcal{L}_{border} = \frac{1}{|\mathcal{D}_t|} \sum_{(x,y) \sim \mathcal{D}_t} \max(0, E(x) - \Theta), \qquad (10)$$

**Midline Loss: Aligning the Central Trend** The midline loss pulls the mean energy of each batch toward a shared reference value $\Delta$, thereby standardizing the central tendency of energy distributions across domains:

$$\mathcal{L}_{midline} = \left| \Delta - \frac{1}{|\mathcal{D}_t|} \sum_{(x,y) \sim \mathcal{D}_t} E(x) \right|, \qquad (11)$$

### 4.3. Hybrid Energy-Distance Weighted Inference Phase

To address Challenge 2, the inference stage adopts an energy–distance weighted domain selection strategy that leverages knowledge from known domains to handle unseen ones. We construct energy and distance factors, each represented as a scalar measuring how well a sample matches a domain; these factors are comparable across domains within a shared scale, enabling relative domain similarity to be inferred by directly comparing their magnitudes.

After incrementally training on $\{\mathcal{D}^1, \ldots, \mathcal{D}^s\}$ and obtaining domain prompts $\{P^1, \ldots, P^s\}$, we process a test image

$x$—whether from a known or unknown domain—as follows. We first extract CLIP image features (without prompts) and compute distances from $x$ to all domain cluster centers, forming distance factors $\{DF^i(x)\}_{i=1}^s$. In parallel, $x$ is passed through each prompt-embedded model $(\mathcal{M}_{pre}, P^i)$ to obtain prediction probabilities $\{P^i(x)\}$ and energy factors $\{EF^i(x)\}$. The energy factors $EF(x)$ and distance factors $DF(x)$ are then fused to compute the hybrid factor $F(x)$:

$$F(x) = \frac{EF(x)}{\alpha} + \frac{DF(x)}{\beta}, \qquad (12)$$

where $\alpha$ and $\beta$ are shared scalar coefficients that balance the contributions of the energy and distance factors, chosen to reduce the search space and mitigate overfitting during hyperparameter tuning.

**Energy and Distance Factor Construction.** The energy factor for the $i$-th domain is denoted as $EF^i(x)$, where $i$ ranges from 1 to $S$. This factor represents the gap between the energy value of the $i$-th domain and the minimum energy value $E_{\min}$, and it serves as a measure of confidence. The energy factor $EF(x)$ ranges from $-\infty$ to 0, where a larger $EF^i$ value corresponds to a higher similarity of the sample to the $i$-th domain, indicating a more confident match. The calculations are as follows:

$$E_{\min} = \min_{i=1}^{S}\{E^i(x)\}, \qquad (13)$$

$$EF^i(x) = E_{\min} - E^i(x), \qquad (14)$$

In pre-trained non-prompt models such as CLIP, features from multiple domains naturally form a shared embedding space with clear clustering. Distances in this space provide a range-consistent scalar measure of domain similarity, allowing direct comparison across domains. This distance factor offers stable prior knowledge and mitigates potential overfitting of energy-based weighting.

Concretely, we use the visual transformer of $\mathcal{M}_{pre}$ (without prompts) to compute domain-specific feature clusters. For domain $i$, image features $\{Z_v(x^{i,j})\}_{j=1}^{|\mathcal{D}^i|}$ are extracted from $\mathcal{D}^i$. Then, K-means partitions them into subsets $\mathcal{D}_j^i$ with cluster centers $M^i = \{M_j^i\}_{j=1}^K$, computed as

$$M_j^i = \frac{1}{|\mathcal{D}_j^i|}\sum_{x \in \mathcal{D}_j^i} Z_v(x) = \frac{1}{|\mathcal{D}_j^i|}\sum_{x \in \mathcal{D}_j^i}\varphi(x|\mathcal{M}_{pre}), \quad (15)$$

where $|\mathcal{D}_j^i|$ denotes the subset size.

For a test sample $x$, the image feature $Z_v(x)$ is first calculated on the pre-trained model $\mathcal{M}_{pre}$. Given the hyperspherical nature of the pre-trained feature space, we employ Cosine distance to calculate the dissimilarity between $Z_v(x)$ and the cluster points $M^i$ of the $i$-th domain. $D^i(x)$ denotes

the minimum distance to the $i$-th domain:

$$D^i(x) = \min_{j=1}^{K}\left(1 - \cos(Z_v(x), M_j^i)\right), \qquad (16)$$

By comparing the distances of a sample to various domains, we can assess the sample's similarity to each domain. Ideally, for a sample $x^i$ belonging to the $i$-th domain, the relationship can be expressed as: $D^i(x^i) < D^j(x^i)$ for all $j \neq i$. Furthermore, the relative distances between the sample $x$ and each domain are calculated to measure distance factors $\{DF^i(x)\}_{i=1}^S$, where $i$ ranges from 1 to $S$, as follows:

$$DF^i(x) = D_{\min} - D^i(x), \qquad (17)$$

where $D_{\min} = \min_{i=1}^{S}\{D^i(x)\}$ is the minimum distance of the sample $x$ to all domains. Clearly, $DF(x) \in (-\infty, 0]$. Within this range, a larger $DF$ value, which corresponds to a smaller $D^i(x)$, means the sample is closer to a domain's clustering points, indicating higher similarity.

**Mixed-Domain Inference.** Next, through Softmax normalization, the weights for all domains $\{W^i\}_{i=1}^S$ are obtained:

$$W^i = \frac{e^{F^i(x)}}{\sum_{j=1}^S e^{F^j(x)}}. \qquad (18)$$

These weights are subsequently applied to the domain-specific prediction probability sets $\{P^i(x)\}_{i=1}^S$, generated by the prompt models for each domain, to obtain the final mixed-domain classification probability $P^{\text{mix}}(x)$:

$$P_{\text{mix}}(x) = \sum_{i=1}^{S} W^i \cdot P^i(x), \qquad (19)$$

where $P_{\text{mix}}(x)$ denotes the aggregated prediction obtained by fusing all domain-specific probabilities according to their respective weights.

## 4.4. Theoretical Analysis

We provide theoretical insights into how HEDP addresses the stability–plasticity dilemma in DIL, with detailed derivations provided in the Appendix.

**Proposition 1. Feature Subspace Orthogonality.** Assuming the frozen backbone extracts semantic-invariant features and prompt tuning models domain-specific statistics, the error gradients of the energy and distance terms with respect to $x$ lie in approximately orthogonal subspaces:

$$\mathbb{E}\left[\langle \nabla_x \mathcal{E}_{EF}(x), \nabla_x \mathcal{E}_{DF}(x)\rangle\right] \approx 0. \qquad (20)$$

**Proposition 2. Regularization Induces Landscape Compactness.** Minimizing $\mathcal{L}_{reg}$ constrains the output range of the energy function, leading to compression of the energy distribution. This compression implicitly suppresses the local Lipschitz constant $K$ of $E(x)$ on the data manifold:

$$|E(x_1) - E(x_2)| \leq K\|x_1 - x_2\|_2. \qquad (21)$$

*Table 1.* DIL Results on CDDB-Hard for both Known and Unknown scenarios. * represents the result is quoted from (Nicolas et al., 2024).

| Method | Prompt | Buffer (/class) | Known AA(↑) | Known AF(↑) | Unknown AA(↑) |
|---|---|---|---|---|---|
| LRCIL | × | | 76.39* | -4.39* | - |
| iCaRL | × | 100 | 79.76* | -8.73* | - |
| LUCIR | × | | 82.53* | -5.34* | - |
| LRCIL | × | | 74.01* | -8.62* | - |
| iCaRL | × | 50 | 73.98* | -14.50* | - |
| LUCIR | × | | 80.77* | -7.85* | - |
| DyTox | ✓ | | 86.21* | -1.55* | - |
| EWC | × | | 50.59* | -42.62* | - |
| LwF | × | | 60.94* | -13.53* | 50.05* |
| DyTox | ✓ | | 51.27* | -45.85* | 50.46* |
| L2P | ✓ | | 61.28* | -9.23* | 57.34* |
| ESN | ✓ | 0 | 53.97 | -19.98 | 55.81 |
| MoP-CLIP | ✓ | | 88.56 | -0.80 | 81.98 |
| S-liPrompts | ✓ | | 88.65 | -0.69 | 78.78 |
| CP-Prompt | ✓ | | 93.65 | -0.25 | 80.4 |
| **HEDP(Ours)** | ✓ | | **93.72** | **-0.08** | **83.74** |

*Table 2.* DIL Results on DomainNet for both Known and Unknown scenarios. * represents the result is quoted from (Feng et al., 2024).

| Method | Prompt | Buffer (/class) | AA(↑) Known(all) |
|---|---|---|---|
| DyTox | ✓ | 50 | 62.94* |
| DyTox | ✓ | | 13.5* |
| EWC | × | | 47.6* |
| LwF | × | | 49.2* |
| SimCLR | × | | 44.2* |
| BYOL | × | | 49.7* |
| Barlow Twins | × | 0 | 48.9* |
| SupCon | × | | 50.9* |
| L2P | ✓ | | 40.1* |
| ESN | ✓ | | 66.46 |
| S-liPrompts | ✓ | | 67.75 |
| MoP-CLIP | ✓ | | 69.72 |
| CP-Prompt | ✓ | | 73.15 |
| **HEDP(Ours)** | ✓ | | **74.19** |

| Method | Prompt | Buffer (/class) | Known(top5) | Unknown |
|---|---|---|---|---|
| ESN | ✓ | | 66.44 | 59.51 |
| S-liPrompts | ✓ | | 67.59 | 58.11 |
| MoP-CLIP | ✓ | 0 | 70.39 | 63.97 |
| CP-Prompt | ✓ | | 73.52 | 57.57 |
| **HEDP(Ours)** | ✓ | | **74.43** | **67.09** |

## 5. Experiments

### 5.1. Experimental setup

**Datasets and Tasks**. To ensure fairness in evaluating the algorithm, we selected three widely used datasets in DIL: CDDB-Hard(Li et al., 2023), DomainNet(Peng et al., 2019), and CORe50(Lomonaco & Maltoni, 2017). We also referenced (Wang et al., 2022a) proposed evaluation metrics: Average Accuracy (AA) and Average Forgetting (AF). The experiments particularly emphasized comparing performance in unknown domain task settings.

**Baselines**. We compare HEDP with state-of-the-art DIL methods. To ensure fairness, we quote results marked with '*' from original papers (Nicolas et al., 2024; Feng et al., 2024) and reproduce others using official implementations under the identical backbone. Non-prompt methods include rehearsal-based (LRCIL (Pellegrini et al., 2020), ER (Chaudhry et al., 2019), DER++ (Buzzega et al., 2020), iCaRL (Rebuffi et al., 2017), LwF (Li & Hoiem, 2016)), regularization-based (LUCIR (Hou et al., 2019), EWC (Kirkpatrick et al., 2017)), and parameter isolation methods (BiC (Wu et al., 2019), GDumb (Prabhu et al., 2020)). We also include Co$^2$L (Cha et al., 2021) and CaSSLe (Fini et al., 2022). Prompt-based competitors include L2P (Wang et al., 2022b), DyTox (Douillard et al., 2022), S-Prompts (Wang et al., 2022a), MoP-CLIP (Nicolas et al., 2024), ESN (Wang et al., 2023b), and CP-Prompt (Feng et al., 2024).

### 5.2. Main Results

**Experiments in Known Domains**. On CDDB-Hard and DomainNet, we conducted the evaluation of the proposed HEDP method for DIL tasks within known domains, where the test samples and training data are drawn from the same distribution. As demonstrated in the "Known" columns of Tables 1 and 2, HEDP exhibited exceptional performance in 2-class and 345-class DIL tasks, achieving the highest improvement reaching 1.04% compared to other methods, including the state-of-the-art domain incremental method CP-Prompt. Additionally, HEDP significantly reduced knowledge forgetting without retaining samples from previous domains, lowering the average forgetting rate to 0.08%. Compared to methods that rely on replay, HEDP not only alleviates storage pressure and potential security risks but also offers superior performance. Experimental results clearly illustrate that HEDP surpasses other prompt-based methods, maintaining high performance and low forgetting rates throughout phased evaluations, which attests to its efficiency in managing old knowledge and acquiring new knowledge.

**Experiments in Unknown Domains**. To validate our method's generalization, we conducted experiments in unknown domains, simulating real-world distribution shifts with datasets not encountered during training. Results in the "Unknown" columns of Tables 1, 2, and 3 fully demonstrate HEDP's superior generalization. HEDP surpassed current state-of-the-art models on CDDB-Hard, DomainNet, and CORe50 by 1.76%, 3.12%, and 2.57%, respectively. Experimental results illustrate that HEDP consistently leads

*Table 3.* DIL Results on CORe50 for Unknown scenarios. * represents the result is quoted from (Feng et al., 2024) .

| Method | Prompt | Buffer (/class) | Unknown $AA(\uparrow)$ |
|--------|--------|-----------------|-------------------------|
| ER | × | | $80.10 \pm 0.56$* |
| GDumb | × | | $74.92 \pm 0.25$* |
| BiC | × | | $79.28 \pm 0.30$* |
| DER++ | × | 50 | $79.70 \pm 0.44$* |
| Co$^2$L | × | | $79.75 \pm 0.84$* |
| DyTox | ✓ | | $79.21 \pm 0.10$* |
| L2P | ✓ | | $81.07 \pm 0.13$* |
| EWC | × | | $74.82 \pm 0.60$* |
| LwF | × | | $75.45 \pm 0.40$* |
| L2P | ✓ | | $78.33 \pm 0.06$* |
| S-liPrompts | ✓ | 0 | $87.07 \pm 0.65$ |
| CP-Prompt | ✓ | | $90.67 \pm 0.55$ |
| MoP-CLIP | ✓ | | $91.43 \pm 0.52$ |
| ESN | ✓ | | $\underline{91.80 \pm 0.31}$ |
| **HEDP(Ours)** | ✓ | | $\mathbf{94.37 \pm 0.29}$ |

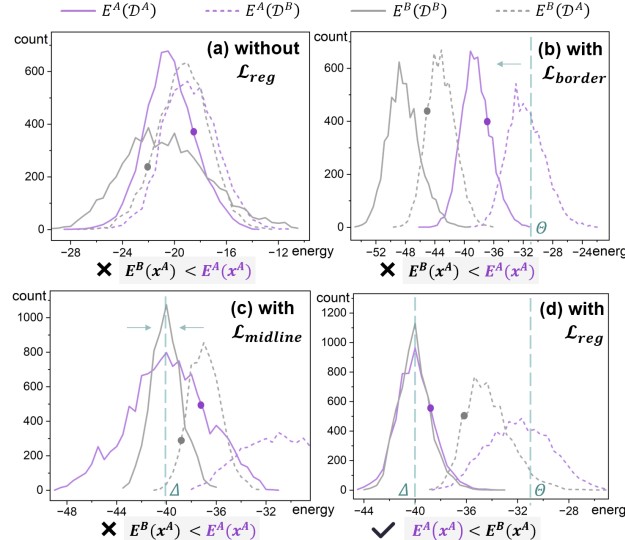

*Figure 4.* The effect of different regularization strategies on the energy statistical distribution between two domains. Subfig(d), utilizing the full $\mathcal{L}_{reg}$, most effectively aligns with the hypothesis that lower energy values correlate with greater domain similarity, surpassing the performance of subfig(a) without $\mathcal{L}_{reg}$, subfig(b) with only $\mathcal{L}_{border}$, and subfig(c) with only $\mathcal{L}_{midline}$. The x-axis measures energy values, and the y-axis indicates their respective counts.

in unknown domain performance throughout incremental updates. Notably, HEDP achieves dual optimization in performance across both known and unknown domains, a feat unachieved by benchmarks like CP-Prompt and MoP-CLIP.

## 5.3. Ablation Study

We designed six ablation schemes (Table 4) to isolate each component's effect on both known and unknown domains. Scheme 1 removed all energy-related terms and relied only on distance factors, leading to notable degradation—especially on unknown domains—revealing distance alone provides weak generalization. Schemes 2–5 removed distance factors to examine energy components. Without energy regularization (Scheme 2), performance on known domains dropped sharply, indicating that regularization is crucial for stable inter-domain similarity and for mitigating forgetting. Adding boundary and midline constraints (Schemes 3 and 4) improved accuracy, and combining them (Scheme 5) yielded further gains, showing that these losses sharpen domain boundaries and enhance similarity estimation. Comparing Scheme 1 and Scheme 5 confirms that distance factors and energy factors are complementary. Scheme 6, which integrates both factors and all regularization terms, achieves the best overall performance by fully leveraging their joint strengths.

Furthermore, to examine the contribution of each component, as shown in Fig.4, we visualize the energy distributions obtained from four combinations of domain data $(\mathcal{D}^A, \mathcal{D}^B)$ and prompts $(P^A, P^B)$, producing four cross-domain energy terms $E^j(x^i)$.

**(a) Without regularization $\mathcal{L}_{reg}$.** Using a single-domain prompt leads to substantial overlap in energy values between in-domain and out-of-domain samples. Even the same input may yield similar energies under different prompts, obscuring prompt–input alignment. for instance, an A-domain sample $x^A$ may satisfy $E^A(x^A) > E^B(x^A)$, misleading the model into treating it as more B-like.

**(b) With $\mathcal{L}_{border}$ only.** Using only the boundary loss pushes in-domain energies below the threshold $\Theta$ to separate domains, yet two issues remain. First, since training never observes out-of-domain samples, their energies under the current prompt are not guaranteed to exceed $\Theta$, allowing cases such as $E^B(x^B) < E^B(x^A) < \Theta$. Second, domains experience uneven downward shifts, further destabilizing cross-domain comparisons; e.g., $E^B(x^A) < E^A(x^A) < \Theta$, causing the model to misclassify $x^A$ as more B-like.

**(c) With $\mathcal{L}_{midline}$ only.** The midline loss aligns each domain's mean energy toward a shared reference $\Delta$, but without boundary constraints certain domains (e.g., B) may become overly compressed while others (e.g., A) remain dispersed, sometimes reintroducing inter-domain overlap and errors such as $E^B(x^A) < E^A(x^A)$.

**(d) With full loss $\mathcal{L}_{reg}$.** The full energy regularization loss $\mathcal{L}_{reg}$—combining boundary and midline terms—avoids both excessive shifts and over-compression, establishing a

*Table 4.* Ablation Results of HEDP on three datasets.

| Scheme | Components | | | | AA($\uparrow$) | | | | | |
|---|---|---|---|---|---|---|---|---|---|---|
| No. | Energy Reg Loss | | Hybrid Factors | | CDDB-Hard | | DomainNet | | | CORe50 |
| | Boundary | Midline | Energy | Distance | Known | Unknown | Known(all) | Known(top5) | Unknown | Unknown |
| 1 | $\times$ | $\times$ | $\times$ | $\checkmark$ | 92.88 | 75.8 | 72.72 | 73.39 | 64.97 | 93.17 |
| 2 | $\times$ | $\times$ | $\checkmark$ | $\times$ | 78.83 | 77.31 | 66.84 | 67.45 | 63.55 | 92.06 |
| 3 | $\checkmark$ | $\times$ | $\checkmark$ | $\times$ | 86.85 | 79.22 | 64.4 | 65.26 | 65.05 | 92.98 |
| 4 | $\times$ | $\checkmark$ | $\checkmark$ | $\times$ | 91.63 | 79.12 | 68.25 | 69.25 | 65.01 | 93.77 |
| 5 | $\checkmark$ | $\checkmark$ | $\checkmark$ | $\times$ | 92.75 | 81.52 | 72.54 | 73.11 | 65.59 | 94.07 |
| **6** | $\checkmark$ | $\checkmark$ | $\checkmark$ | $\checkmark$ | **93.72** | **83.74** | **74.19** | **74.43** | **67.09** | **94.66** |

*Figure 5.* Heatmap of HEDP's performance with $\alpha$ and $\beta$ from 0.1 to 1. Axes represent $\alpha$ and $\beta$, respectively, and colors indicate average accuracy.

unified energy scale across prompts. When $E^A(x^A) \approx E^B(x^B)$ and $E^B(x^B) < E^B(x^A)$, it follows that $E^A(x^A) < E^B(x^A)$, enabling reliable identification of the domain most compatible with a given sample.

### 5.4. Energy Midline, Boundary, and Clustering Effects

We analyze the influence of these hyperparameters under unknown domain shifts on CDDB, DomainNet, and Core50, as illustrated in Figure 6 in the appendix, and summarize our conclusions as follows.

**(1) Energy Midline** $\Delta$**.** Pushing $\Delta$ away from zero consistently boosts accuracy until reaching a stable plateau, indicating that a well-spaced midline expands inter-domain energy separation and strengthens the energy factor's ability to discriminate between intra- and inter-domain samples.

**(2) Energy Boundary** $\Theta$**.** Optimal performance emerges when $\Theta$ maintains a proper margin from $\Delta$, yielding a stable energy structure and clearer inter-domain distinctions. This improves domain-similarity estimation and cross-domain robustness. When $\Theta$ is placed too close to or too far from $\Delta$, the discriminative structure collapses and accuracy declines.

**(3) Number of Clusters** $K$**.** Although varying $K$ causes visible fluctuations, the overall gains are minor. This reflects the limitations of distance-only representations: the distance factor struggles to capture deeper cross-domain structural

differences and thus cannot deliver strong generalization in complex DIL settings.

### 5.5. Effects of Energy and Distance Scaling Factors

*Scaling Factors $\alpha$ and $\beta$ in Fig. 5*: In Scheme 6, we fine-tuned the model using a grid search for $\alpha \in [0.1, 1]$ and $\beta \in [0.1, 1]$. The heatmaps reveal distinct operational mechanisms for known versus unknown domains.On **known domains** (Fig. 5(a) and (b)), performance improves in the lower triangular regions where distance factors are prioritized (smaller $\beta$). This indicates that the distance metric, derived from the frozen backbone's stable embedding space, is crucial for preserving historical knowledge and mitigating catastrophic forgetting.Conversely, for **unknown domains** (Fig. 5(c)), accuracy peaks near the diagonal equilibrium. Extreme imbalances (e.g., $\alpha = 0.1$ or $\beta = 0.1$) cause significant degradation. This suggests that generalization relies on the synergy between the global topology provided by distance factors and the local discriminability of energy factors. The optimal equilibrium highlights that factor complementarity is essential for robust open-world adaptability.

## 6. Dissusion

**Why is the energy term necessary beyond distance-only selection?** The proposed hybrid domain selection mechanism is motivated by the limitation of distance-only se-

lection under ambiguous domain boundaries. The distance factor provides a stable geometric criterion in the CLIP feature space, but it can be unreliable when cluster centers from different domains are close to each other or when the test sample lies near fuzzy inter-domain boundaries. In such cases, a sample from the true domain $s$ may have a smaller distance to another domain $j$, i.e., $D_j < D_s$, leading to an incorrect domain assignment if only the distance term is used.

The energy term serves as an explicit correction signal because it measures the compatibility between the input sample and each domain-specific prompt model. Let the hybrid selection score for domain $i$ be defined as

$$\mathcal{J}_i(x) = \alpha D_i(x) + \beta E_i(x), \tag{22}$$

where $D_i(x)$ denotes the distance factor, $E_i(x)$ denotes the energy factor, and $\alpha, \beta$ control their relative contributions. For a sample $x$ from the true domain $s$, the correct domain can still be selected over a competing domain $j$ as long as

$$\mathcal{J}_s(x) < \mathcal{J}_j(x). \tag{23}$$

This condition is equivalent to

$$E_j(x) - E_s(x) > \frac{\alpha}{\beta}\big(D_s(x) - D_j(x)\big). \tag{24}$$

Therefore, even when the distance term incorrectly favors domain $j$, the final decision remains correct if the energy gap between the incorrect domain and the true domain is sufficiently large. This shows that the energy term is not redundant with the distance term. Instead, it provides a complementary prompt-conditioned statistical criterion that corrects distance-based failures caused by geometrically ambiguous samples.

**Why can shared $\Theta/\Delta$ reduce energy overlap?** Regarding the ranking issue, during training, we apply energy regularization to enforce separation among domain models. Consequently, lower energy directly indicates higher similarity, enabling domain selection during inference.

For different intervals for each domain: Since the visual distribution is continuous and, after model transformation and output, becomes a near-normal distribution, the distribution of each domain spreads throughout the entire space. Therefore, it's impossible to completely divide the distribution of each domain into different intervals. Thus, we use boundary and midline losses to improve the orthogonality of the domain expert models.

For reducing domain overlap. In essence, the two loss terms jointly perform center alignment and boundary compression on the approximately Gaussian-like energy distributions induced by different prompts, thereby shifting them apart and reducing inter-domain overlap.

**Computational complexity of HEDP. Training Phase.** Learning a new domain requires no rehearsal of past data or previous parameters. Thus, the time complexity for training a single new domain is $\mathcal{O}(1)$ with respect to the number of domains $S$, ensuring excellent scalability.

**Inference Phase.**

- Distance Factor. CLIP feature extraction ($\mathcal{O}(C_{vit})$) and cosine distance to $K$ cluster centers across $S$ domains ($\mathcal{O}(K \cdot S \cdot d)$). Total: $\mathcal{O}(C_{vit} + K \cdot S \cdot d)$.

- Energy Factor & Prediction. Forward passes through $S$ prompted models to obtain predictions and EFs. Complexity: $\mathcal{O}(S \cdot C_{vit})$.

- Hybrid Weight Fusion. Calculating soft weights and aggregating predictions across $S$ domains. Complexity: $\mathcal{O}(S \cdot U)$, where $U$ is the number of categories.

**Total Inference.** Summing these steps and omitting lower-order terms, the asymptotic time complexity is dominated by the Transformer passes, resulting in $\mathcal{O}(S \cdot C_{vit})$, which scales linearly as $\mathcal{O}(S)$.

## 7. Limitations and Future Work

HEDP's inference latency scales linearly with the number of domains, posing a scalability challenge. Future work will investigate dynamic prompt selection to reduce computational overhead and adaptive hyperparameter calibration to further improve open-world robustness.

## 8. Conclusion

This paper presents HEDP, a prompt-based framework for domain incremental learning that effectively prevents catastrophic forgetting and generalizes to unknown domains without replaying data. HEDP introduces an energy regularization loss to transform energy into a reliable domain similarity metric while preserving classification accuracy. HEDP outperforms existing methods on three benchmarks, proving its effectiveness in both known and unknown domains.

## Acknowledgement

This work is supported by the National Key Research and Development Program of China under Grant 2024YFC3308500, Beijing Municipal Natural Science Foundation under Grant L251042, National Natural Science Foundation of China under Grant 62406036, China Postdoctoral Science Foundation under Grant 2025M781457, and also sponsored by the State Key Laboratory of Networking and Switching Technology under Grant NST20250110.

## Impact Statement

This research contributes to the advancement of machine learning with a specific focus on Domain Incremental Learning. While the broader societal implications of such foundational work are multifaceted, we do not foresee any immediate negative consequences that require specific highlighting. Our experiments rely exclusively on publicly available benchmarks that are free of sensitive or personally identifiable information. By enhancing Hybrid Energy-Distance Prompt technology, this work aims to drive positive scientific progress without raising ethical or privacy concerns.

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

# Appendix A. Theoretical Derivations

In this section, we elaborate on the theoretical motivations presented in Section 4.4, demonstrating how HEDP mitigates domain shifts through error decoupling and landscape smoothing.

## A.1. Analysis of Decoupled Error Modes (Proposition 1)

**Motivation:** A key challenge in prompt-based DIL is the sensitivity of prompts to domain shifts. We posit that HEDP improves robustness because it combines two fundamentally different views of the data, defined as the Distance Factor ($DF$) and Energy Factor ($EF$):

- **Distance Factor ($DF$):** Derived from the frozen CLIP backbone. It captures *global, semantic-invariant features* (e.g., object shapes) but lacks domain-specific discriminability. Its error mode is primarily semantic ambiguity.

- **Energy Factor ($EF$):** Derived from domain-specific prompts. It captures *local, distribution-specific nuances* (e.g., texture, lighting style). Its error mode is sensitivity to statistical distribution shifts.

**Derivation:** Let $\mathcal{E}_{EF}$ and $\mathcal{E}_{DF}$ denote the estimation errors of the Energy Factor and the Distance Factor on an unseen domain sample $x_{out}$. The hybrid factor is defined as $F(x) \propto \alpha \cdot EF(x) + \beta \cdot DF(x)$. The Mean Squared Error (MSE) of this hybrid estimator on an unseen domain contains a covariance term $\Omega_{cov}$ between the errors of the two factors.

Unlike standard ensembles where base learners share similar biases, HEDP exploits the structural difference in gradients:

1. The gradient $\nabla_x DF(x)$ lies in the subspace of **semantic features**, dictated by the pre-trained manifold of $M_{pre}$.

2. The gradient $\nabla_x EF(x)$ is dominated by the prompt parameters $P$, which act as high-frequency modulations targeting **domain statistics**.

Since semantic content and domain style (texture/noise) typically occupy different subspaces in the high-dimensional input representation, the directions of maximum sensitivity for $DF$ and $EF$ are distinct. Mathematically, we approximate the error covariance as:

$$Cov(\mathcal{E}_{EF}, \mathcal{E}_{DF}) \approx \langle \nabla_x \mathcal{E}_{EF}, \nabla_x \mathcal{E}_{DF} \rangle \approx 0. \quad (25)$$

Consequently, the hybrid mechanism effectively filters out uncorrelated errors: a perturbation that confuses the distance metric is unlikely to simultaneously trigger a high-confidence response in the energy model, and vice versa.

## A.2. Proof of Energy Landscape Smoothing (Proposition 2)

**Motivation:** In standard training, decision boundaries can become arbitrarily sharp, leading to "energy collapse" where an out-of-distribution (OOD) sample close to the boundary is incorrectly assigned low energy. We show that $\mathcal{L}_{reg}$ mitigates this by enforcing output compactness.

**Derivation:** Recall the energy regularization loss (with $\mathcal{L}_{border}$ defined as a penalty on excess energy):

$$\mathcal{L}_{reg} = \mathcal{L}_{border} + \mathcal{L}_{midline}$$
$$= \frac{1}{|\mathcal{D}|} \sum \max(0, E(x) - \Theta) + \left| \Delta - \frac{1}{|\mathcal{D}|} \sum E(x) \right|. \quad (26)$$

Minimizing $\mathcal{L}_{midline}$ centers the energy distribution at $\Delta$, while minimizing $\mathcal{L}_{border}$ strictly penalizes any energy values exceeding $\Theta$. This effectively compresses the energy responses of in-domain samples into a compact interval $(-\infty, \Theta]$ clustered around $\Delta$.

Consider the local behavior of the energy function $E(x)$ via its first-order Taylor expansion around a sample $x_0$:

$$E(x) \approx E(x_0) + \nabla_x E(x_0)^T (x - x_0). \quad (27)$$

While $\mathcal{L}_{reg}$ does not explicitly bound the gradient norm $\|\nabla_x E(x)\|$, it imposes a severe constraint on the output variance. For a neural network with bounded weights (ensured via weight decay), restricting the output variation $|E(x) - \Delta|$ to be small for all training samples implicitly discourages high-frequency oscillations or sharp gradients on the data manifold.

**Compactness Implies Stability.** By forcing the energy surface to be "flat" (values close to $\Delta$) throughout the in-domain region, we implicitly limit the local Lipschitz constant $K$. For an unknown domain sample $x_{out} = x_{in} + \Delta$, the deviation is bounded by:

$$|E(x_{out}) - E(x_{in})| \leq K\|\Delta\|. \quad (28)$$

By compressing the output space via $\mathcal{L}_{reg}$, HEDP minimizes $K$, preventing abrupt energy drops for OOD samples. This creates a "safe buffer," ensuring that unknown domains maintain higher energy values relative to the known domain, thereby reducing catastrophic forgetting.

# Appendix B. Experimental details

**Datasets and Tasks**. DIL tasks are designed to assess algorithm, focusing on preventing catastrophic forgetting in known domains while enhancing generalization to unknown domains. We have selected three well-known datasets, each featuring multiple domains with different data distributions, divided into known and unknown sets. Upon completing

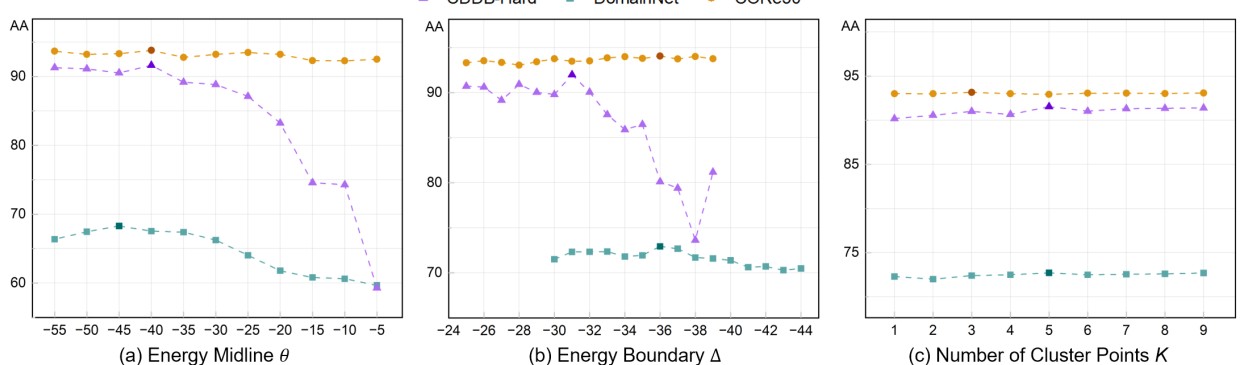

*Figure 6.* Hyper-parameter tuning of HEDP: $\Theta$, $\Delta$ and $K$. The x-axis shows parameter values, while the y-axis represents average accuracy.

training on a domain, we assess the model's performance on both the current known and unknown domains before proceeding to the next domain, without replaying samples from prior domains. Below is the explanation of the datasets and their division:

CDDB (Li et al., 2023) is a deepfake detection dataset, where fake images are generated from real ones using diverse generative models, forming domain shifts for authenticity classification. We adopt the Hard setting, which includes five known domains—GauGAN, BigGAN, WildDeepfake, WhichFaceReal, and SAN—and three unseen domains: GLOW, StarGAN, and CycleGAN.

DomainNet (Peng et al., 2019) is a large-scale image dataset comprising six stylistically diverse domains—Clipart, Infograph, Painting, Quickdraw, Real, and Sketch—each containing 345 categories. We first evaluate all six domains under the DIL setting. We then train on the first five domains and treat the remaining one as an unseen target domain for evaluation.

CORe50 (Lomonaco & Maltoni, 2017) is an object recognition dataset with 50 categories, featuring images captured under varying conditions across 11 domains, including 8 indoor and 3 outdoor domains. We train on the indoor domains and evaluate generalization on the outdoor domains.

**Evaluation metrics**. Evaluation metrics. We adopt standard evaluation metrics from the DIL literature (Wang et al., 2022a). Average Accuracy ($AA$) measures inter-domain performance, reflecting incremental learning ability and resistance to catastrophic forgetting. Average Forgetting ($AF$) quantifies the performance degradation on previously learned domains. For CORe50, $AF$ is not reported due to the inherent discrepancies between indoor and outdoor domains. The definitions of $AA$ and $AF$ are given below, where $A_{i,j}$ denotes the accuracy on the $i$-th domain after training on the $j$-th domain.

$$AA = \frac{1}{S} \sum_{i=1}^{S} A_{i,S}, \tag{29}$$

$$AF = \frac{1}{S-1} \sum_{i=1}^{S-1} (A_{i,S} - A_{i,i}), \tag{30}$$

where $A_{i,i}$ represents the initial accuracy of domain $i$ immediately after it was learned, and $A_{i,S}$ is its accuracy after the final domain $S$ is learned.

**Implementation Details**. We implemented the HEDP method using a **frozen CLIP ViT-B/16** backbone on an NVIDIA GeForce RTX 3090 GPU with Python 3.8.12 and PyTorch 1.11.0. The SGD optimizer was set with a cosine annealing scheduler, an initial learning rate of 0.01, momentum at 0.9, and weight decay at 0.0005. Training epochs were 50 for CDDB, 30 for DomainNet, and 20 for CORe50, with a batch size of 128 across all experiments. The energy regularization loss ratio $\lambda$ was set to 0.05, with Boltzmann constants $k$ and $T$ at 1.

**Hyperparameter Selection**. We conduct a comprehensive sensitivity analysis of the key hyperparameters across three tasks. Based on the experimental results, we identify stable empirical ranges of $\alpha \in [0.7, 0.9]$ and $\beta \in [0.4, 0.6]$, within which the model consistently achieves strong performance. We further evaluate extreme settings, e.g., when the parameter value is close to $0.1$, and observe a moderate performance degradation. Notably, the decrease is limited to approximately 2–3 percentage points rather than a sharp or catastrophic drop. This indicates that the proposed method is not overly sensitive to the exact hyperparameter values and maintains robust performance under a relatively broad range of settings, which supports its practicality in deployment.

