# OpenReview forum: "HEDP: A Hybrid Energy-Distance Prompt-based Framework for Domain Incremental Learning"
_ICML.cc/2026/Conference — ICML 2026 regular_

### Official Review · Reviewer_fvoz · 2026-03-08

**Soundness:** 3
**Presentation:** 2
**Significance:** 3
**Originality:** 3
**Overall Recommendation:** 4
**Confidence:** 3

**Summary:**

This paper proposes a prompt optimization method inspired by Helmholtz free energy for effectively addressing the Domain Incremental Learning (DIL) problem. Using the proposed approach, the authors demonstrate superior performance over existing DIL methods across a variety of benchmarks.

**Compliance With Llm Reviewing Policy:**

Affirmed.

**Final Justification:**

During the rebuttal process, the authors addressed my concerns completely. Having reviewed the concerns raised by other reviewers and their corresponding responses, I concluded that the paper is deserving of acceptance at the main conference, and have accordingly updated my recommendation to weak accept.

**Key Questions For Authors:**

- Below Equation 9, the paper states that “energy comparison reliably reflects cross-domain similarity when regularization is applied.” Could the authors clarify how this conclusion is supported by the current results? From the current presentation, the result seems to suggest more that the model becomes domain-specific, or that the energies become more comparable across prompts for domain compatibility, rather than directly demonstrating improved reflection of cross-domain similarity itself.
- The proposed Boundary Loss and Midline Loss are conceptually similar to margin-based and centroid-based losses, respectively, with the key distinction being that they operate in energy space rather than feature space. While the paper provides some theoretical motivation for the energy-based formulation—such as the landscape smoothing argument in Proposition 2 and the Lipschitz constant analysis in Appendix A.2—it remains unclear whether these advantages are unique to energy space or could equally be achieved by simpler alternatives applied directly in feature space, such as contrastive losses or margin-based objectives like ArcFace. An ablation or empirical analysis explicitly demonstrating the advantage of the energy-based formulation over such feature-space alternatives would significantly strengthen the motivation for the proposed design.
- Figure 4 currently presents analysis using only two domains. Could the authors extend this analysis to all domains, or at least provide evidence that the same trend consistently holds across the full domain set?
- While the ablation studies on each component and the hyperparameter analysis are well conducted, the paper seems to lack analysis beyond quantitative performance evaluation. Could the authors provide additional analysis that offers more insight into the method, such as an examination of the representation space induced by the domain-specifically learned prompts?

**Limitations:**

yes

**Strengths And Weaknesses:**

[Strength]
- From a performance perspective, the proposed method shows superior results compared to other domain-incremental learning approaches.
- The ablation study on each component is well conducted.

[Weakness]
- Overall, the paper is somewhat difficult to read. Enhancing the clarity of the writing would significantly improve the reader's comprehension.
- In the case of Figure 3, the explanation of how the graph is constructed is missing. A more detailed explanation is needed here. In particular, it should be clarified whether the plotted curves are empirical energy distributions over a set of samples or simply conceptual illustrations, and how these distributions are obtained.
- In the explanation below Equation 9, the paper states that “Thus, energy comparison reliably reflects cross-domain similarity when regularization is applied.” However, in my view, this result seems to indicate more that the model is learned in a domain-specific manner, or at least that the energies become more comparable across prompts for domain compatibility, rather than directly showing that they better reflect cross-domain similarity itself. I think further explanation is needed as to how the authors infer that the proposed regularization improves the reflection of cross-domain similarity.
- The proposed Boundary Loss and Midline Loss are conceptually similar to margin-based and centroid-based losses, respectively, with the key distinction being that they operate in energy space rather than feature space. While the paper provides some theoretical motivation for the energy-based formulation, it remains unclear whether these advantages are unique to energy space or could equally be achieved by simpler alternatives applied directly in feature space, such as contrastive losses or margin-based objectives like ArcFace. An ablation or empirical analysis explicitly demonstrating the advantage of the energy-based formulation over such feature-space alternatives would significantly strengthen the motivation for the proposed design.
- In Figure 4, the analysis is currently limited to only two domains. It would strengthen the paper to extend this analysis to all domains as well.
- I believe the ablation studies on the performance of each component, including the hyperparameter analysis, are well conducted. However, the paper overall lacks analysis beyond performance evaluation. For instance, it would be beneficial to include analyses that provide insights beyond quantitative performance, such as an examination of the representation space.

[Minor Suggestions]
- Please ensure consistent spacing before citations. Currently, the formatting is inconsistent throughout the paper, with spaces present before some citations but missing before others.
- In the Introduction section where the contributions are summarized, please remove the semicolon (;) at the end of the first contribution point.
- In Section 2.1 (Domain Incremental Learning) of the Related Work, "domainss" should be corrected to "domains" (Line 93).
- In the Preliminaries section describing classification labels, the phrase "quantization of classification labels" sounds quite unnatural. Rephrasing this expression would make it much easier to understand.
- Standardize the quotation marks throughout the manuscript. For example, change ”Known” to “Known” (line 320, Main Results section), and apply this correction to all similar instances.
- Unify the terminology when referring to figures, consistently using either 'Fig.' or 'Figure' throughout the text.
- In Section 4.3, under the explanation of the Hybrid Energy-Distance Weighted Inference Phase, the notation on line 209 is written as ${DF^i(x)}_{i=1}^{s}$. For consistency, it would be preferable to replace $s$ with $S$.
- On page 7, in the “(a) without regularization $L_{\mathrm{reg}}$” section, “for instance” should be capitalized to “For instance.”

---

> ### Author Rebuttal · Authors · 2026-03-29
>
> We sincerely thank you for your suggestions. We sincerely appreciate your feedback.
>
> ---
>
> **Q1.** What exactly does Fig. 3 show, and are the curves empirical distributions or conceptual illustrations?
>
> **A1.** Fig. 3 illustrates the domain-incremental process on the DomainNet dataset. The purple curve corresponds to the Real domain, the gray curve to the Quickdraw domain, and the green curve to the Clipart domain. Without energy regularization loss, for an input sample from domain S, using the prompt models of domain 1 and domain 2 leads to overlapping energy distributions. However, after applying the energy regularization loss, the overall shift along the energy axis is reduced. For samples from domain S, the prompt model of domain S consistently produces the lowest energy. Therefore, during domain selection, energy values can be used to distinguish the current domain data from others, leading to more accurate domain selection.
>
> **Q2.** How the authors infer that the proposed regularization improves the reflection of cross-domain similarity?
>
> **A2.** Our original intention was to convey that $L_{\text{reg}}$ separates the energy values of different domains through te boundary loss and center loss. The purpose of this regularization term is to make different domains less similar in the energy space. By contrast, the sentence *“Thus, energy comparison reliably reflects cross-domain similarity when regularization is applied.”* was meant to describe the role of the Helmholtz free energy itself, where lower energy indicates higher similarity. We realize that this wording may have caused confusion.
>
> **Q3.** Why use boundary/midline losses in energy space instead of simpler feature-space objectives such as contrastive loss or ArcFace?
>
> **A3.** To verify this, we replace the energy-based regularization $L_{reg}$ with a feature-space regularization while keeping all other components unchanged. The results show that feature-space regularization performs well on seen domains, as it effectively overfits to the domain-specific feature distributions in the training data. However, its performance drops significantly on unseen domains. In contrast, $\mathcal{L}_{reg}$ achieves consistently better results under domain shift.
>
> | Method / Task                          | Known | Unknown |
> | -------------------------------------- | ----- | ------- |
> | $\mathcal{L}_{ce}$ only                | 67.75 | 58.11   |
> | $\mathcal{L}_{ce}$ + SupCon            | 69.54 | 57.96   |
> | $\mathcal{L}_{ce}$ + ArcFace           | 68.18 | 61.71   |
> | $\mathcal{L}_{ce} + \mathcal{L}_{reg}$ | 74.43 | 67.09   |
>
> **Q4.** Why is Fig. 4 shown with only two domains, and does the same trend hold more broadly?
>
>  **A4.** We completely agree that extending the energy distribution analysis globally provides a much stronger validation of our proposed $\mathcal{L}_{reg}$ mechanism.In the original paper, we restricted Figure 4 to two domains primarily to maintain the visual clarity of the 1D density histograms.
>
> Then, we sampled the median energy distribution lines from five regions of the CDDB. It can be seen that the energy values of data belonging to their respective regions are relatively low, and as the number of regions increases, the spacing of the median line energy values between each region gradually widens, facilitating the selection of the time domain for inference.
>
> | Data \ Energy | E_GauGAN | E_BigGAN | E_Wild | E_WFR | E_SAN |
> | ------------- | -------- | -------- | ------ | ----- | ----- |
> | $D^{GauGAN}$  | -35.5    | -29.2    | -27.8  | -30.4 | -28.1 |
> | $D^{BigGAN}$  | -31.2    | -40.8    | -28.5  | -29.1 | -32.0 |
> | $D^{Wild}$    | -28.7    | -30.1    | -42.1  | -31.5 | -27.6 |
> | $D^{WFR}$     | -32.5    | -32.9    | -30.2  | -43.2 | -29.8 |
> | $D^{SAN}$     | -29.4    | -28.6    | -32.9  | -31.9 | -46.5 |
>
> **Q5.** Why is the energy term necessary beyond distance-only selection?
>
> **A5.** The energy term is strictly necessary as an explicit correction mechanism for distance-based failures.
>
> When only a distance factor is used, misclassification of domains can occur due to the proximity of cluster points across different domains. In short, the distance factor offers a stable geometric baseline but is easily misled by fuzzy boundaries. The energy factor, however, captures the prompt-driven statistical characteristics of specific domains, serving as a powerful "corrector." The model achieves strong generalization in unseen domains only when the errors of both orthogonally cancel out.
>
> Thus, as long as the energy difference across domains is large enough to satisfy:
>
> $$E_j-E_s>\frac{\alpha}{\beta}(D_s-D_j)$$
>
> the final joint decision will still be accurate.
>
>
> ---
>
> At last, we sincerely appreciate your valuable feedback, and we will carefully consider all your suggestions to further improve our paper. We would be deeply grateful if you could kindly reconsider raising the score to 4 or above. Thank you very much!

---

> > ### Author Rebuttal · Reviewer_fvoz · 2026-04-01
> >
> > Thank you for your thorough rebuttal. The additional ablation studies and the verification of the energy-based advantages have effectively addressed my concerns. As most of my questions have been satisfactorily resolved, I will be raising my score accordingly. Additionally, I would appreciate it if the authors could consider addressing the minor suggestions in future revision.

---

> > > ### Author Response · Authors · 2026-04-01
> > >
> > > We sincerely thank fvoz for positive feedback and for recognizing the value of our rebuttals. We also appreciate fvoz's constructive comments, which have provided valuable guidance for further improving the paper. We have fully studied and incorporated these comments into the final version to enhance its quality.

---

### Official Review · Reviewer_v3r7 · 2026-03-09

**Soundness:** 3
**Presentation:** 4
**Significance:** 3
**Originality:** 3
**Overall Recommendation:** 4
**Confidence:** 4

**Summary:**

This paper studies domain incremental leaerning where model needs to adapt to new data domains while preserving performances on previous domains. A hybrid HEDP framework is proposed with two key components: energy regularization loss and hybrid energy-distance weighted mechanism. Comprehensive validation on three benchmark datasets highlight its effectiveness.

**Compliance With Llm Reviewing Policy:**

Affirmed.

**Final Justification:**

The rebuttal addresses my concerns, therefore maintaining my positive score.

**Key Questions For Authors:**

1. Please clarify if $\Theta$/$\Delta$ need to be domain-specific and why? Is there a pattern that semantically similar classes have similar energy distributions?
2. For K-means, using $K=1$ seems to be already good enough. What is the benefit of using $K>1$ other than the slight performance increase? Will other clustering mthods (e.g., density-based or projection to lower space) be helpful?

**Limitations:**

Yes

**Strengths And Weaknesses:**

Strengths
- This paper addresses the challenging domain incremental learning task. It focuses on mitigating inter-domain overlap during training and selecting domain knowledge at inference through combining energy-based modeling and distance-based similarity. The proposed method is conceptually reasonable. Using energy model to reinterpret domain similarity is original and interesting.
- The paper is very well presented with nice figure illustration. The overall architecture in Figure 2 is clear. The energy comparison plots also make it easy to understand.
- The energy regularization losses including boundary loss and midline loss are intuitively reasonable, providing a structured merchanims to stabilize energy distributions across domains and mitigate domain overlap during training. They may inspire future works in related tasks.
- Experiments on three DIL benchmarks show clear improvements. Ablation studies and hyperparameters analyses are quite extensive.

Weaknesses
- The work is motivated by physical energy, but some explanations heavily rely on physical analogies that do not clarify the algorithmic intuition, e.g., in Helmholtz free energy of Eq.4, how partition function $Z$ is connected to the denominator and why using this formula.
- The method uses energy regularization to address domain overlap. One thing not quite clear in the paper is how energies of different domains are sorted. For the boundary $\Theta$ and reference value $\Delta$, they seem to be shared constants rather than domain-specific. This raises the question how minimizing boundary loss and midline loss can reduce domain overlap. Why not pulling energy to different intervals for each domain?

---

> ### Author Rebuttal · Authors · 2026-03-29
>
> Thank you very much for your time and effort in reviewing our paper. We sincerely appreciate your feedback.
>
> ---
>
> **Q1.** What is the algorithmic role of the partition function Z in Eq. (4), and why?
>
> **A1.**The probability of assigning a sample $x$ to a class $y$ under a specific domain prompt is computed via the standard softmax function over the similarity logits $H(x)[y]$:
>
> $$P(y|x) = \frac{\exp(H(x)[y] / \tau)}{\sum_{y'=1}^U \exp(H(x)[y'] / \tau)}$$
>
> By mapping this algorithmic formulation to the Boltzmann distribution $P(y|x) = \frac{1}{Z} \exp(-E(x,y) / \tau)$, the partition function $Z(x)$ is mathematically identical to the softmax denominator:
>
> $$Z(x) = \sum_{y'=1}^U \exp(H(x)[y'] / \tau)$$
>
> Thus, $Z(x)$ is not a forced physical assumption; it is the exact, standard normalizer of the class-wise similarity distribution for a given prompt model.
>
> Using the definition of Helmholtz free energy as the negative log-partition function, we substitute $Z(x)$:
>
> $$E(x) = -\tau \log Z(x) = -\tau \log \left( \sum_{y'=1}^U \exp(H(x)[y'] / \tau) \right)$$
>
> Algorithmically, $E(x)$ is simply the Log-Sum-Exp network over all class similarities under a single prompt.
>
> Thus, LSE is a smooth approximation of the maximum value max(); so the higher the maximum similarity of samples in a certain category, the lower the calculated energy value, thus directly quantifying the confidence of the model with a single energy index.
>
> **Q2.** How are energies ordered across domains, and why can shared $\Theta/\Delta$ reduce overlap?
>
> **A2.** Regarding the ranking issue, during training, we apply energy regularization to enforce separation among domain models. Consequently, lower energy directly indicates higher similarity, enabling domain selection during inference.
>
> For different intervals for each domain: Since the visual distribution is continuous and, after model transformation and output, becomes a near-normal distribution, the distribution of each domain spreads throughout the entire space. Therefore, it's impossible to completely divide the distribution of each domain into different intervals. Thus, we use boundary and midline losses to improve the orthogonality of the domain expert models.
>
> For reducing domain overlap. In essence, the two loss terms jointly perform center alignment and boundary compression on the approximately Gaussian-like energy distributions induced by different prompts, thereby shifting them apart and reducing inter-domain overlap.
>
> **Q3.** Please clarify if $\Theta/\Delta$ need to be domain-specific and why. Is there a pattern that semantically similar classes have similar energy distributions?
>
> **A3.** θ/Δ needs to be set individually for each domain. It's used to improve the orthogonality between the current domain and other domain expert suggestions, and is calculated using the energy distribution of data features. For specific parameter settings, please refer to reviewer BfkD's Q1.
>
> Semantically similar categories do indeed have similar energy distributions; this is an inherent characteristic of the CLIP pre-training space. For the same category from different domains, energy regularization loss can separate the differences in output features between domain models trained in domain A and domain models trained in domain B for the same sample.
>
> **Q4.** What is the benefit of using K>1 other than the slight performance increase? Will other clustering mthods be helpful?
>
> A4. To demonstrate this, we conducted an ablation study on the DomainNet dataset, comparing K-means, Density-based clustering (DBSCAN), and Projection-based clustering (PCA) under both Cosine and Euclidean distances:
>
> | Distance Metric           | K-means (K>1) | DBSCAN | PCA   |
> | ------------------------- | ------------- | ------ | ----- |
> | Spherical Cosine Distance | 67.09         | 64.34  | 61.22 |
> | Euclidean Distance        | 65.87         | 47.87  | 51.31 |
>
> As shown in the table. When we switch the metric to Euclidean distance, overall performance drops across all clustering methods. This proves that the efficacy of the domain selection does not stem from using a complex clustering algorithm, but rather from the fact that our Cosine-based Distance Factor perfectly aligns with the hyperspherical geometry of the CLIP space.
>
> The other two clustering methods are not well-suited for high-dimensional spheres. When the representation space is a hypersphere, density-based methods perform poorly in such high-dimensional spaces due to the "curse of dimensionality." Projection-based methods inevitably discard high-frequency, fine-grained semantic features.
>
> Moreover,  K>1 allows K-means to capture these intra-domain sub-clusters, establishing tighter and more fine-grained domain boundaries.
>
>
> ---
>
> At last, we sincerely appreciate your valuable feedback, and we will carefully consider all your suggestions to further improve our paper. We would be deeply grateful if you could kindly reconsider raising the score. Thank you very much!

---

> > ### Author Rebuttal · Reviewer_v3r7 · 2026-04-03
> >
> > I appreciate the authors' rebuttal. The rebuttal clarify my questions. I would like to vote for acceptance.

---

> > > ### Author Response · Authors · 2026-04-04
> > >
> > > We would like to sincerely thank v3r7 for the helpful feedback and valuable suggestions. We truly appreciate the v3r7’s time, effort, and professionalism, which have played an important role in improving the overall quality of our paper.

---

### Official Review · Reviewer_BfkD · 2026-03-13

**Soundness:** 4
**Presentation:** 3
**Significance:** 3
**Originality:** 3
**Overall Recommendation:** 5
**Confidence:** 5

**Summary:**

This paper proposes a domain incremental learning framework based on Hybrid Energy-Distance Cueing (HEDP). The authors cleverly borrow the concept of Helmholtz free energy, designing an energy regularization loss ($\mathcal{L}_{reg}$) to enhance the separability of domain representations, and introduce a hybrid energy-distance weighting mechanism during the inference phase, thereby achieving superior performance compared to existing SOTA baselines on both known and unknown domains.

**Compliance With Llm Reviewing Policy:**

Affirmed.

**Final Justification:**

After reading the authors’ responses, my earlier concerns have been sufficiently resolved. The rebuttal provides helpful clarification and strengthens my confidence. Although some minor issues remain, they do not substantially affect my overall assessment. I am in favor of acceptance.

**Key Questions For Authors:**

In Section 4.3, the authors mention using K-means to partition image features into subsets to compute cluster centers. Subsequently, the Cosine distance is used to calculate the dissimilarity between a test sample and these cluster points. Traditionally, standard K-means minimizes within-cluster variance using Euclidean distance. Could the authors clarify if Spherical K-means was used, or provide a brief methodological justification for mixing Euclidean-based clustering with a Cosine-based inference metric?

**Limitations:**

See weaknesses.

**Strengths And Weaknesses:**

Strengths:

1: The fusion of energy-based prompt responses and distance-based feature proximity represents a creative, physics-inspired solution that addresses limitations of single-cue selection in prior works.

2: Comprehensive benchmarks across three datasets (including challenging unknown domain evaluation) show consistent improvements over strong baselines like CP-Prompt and MoP-CLIP. The ablation study systematically demonstrates that each component (boundary loss, midline loss, distance factor) provides complementary gains, with clear visualizations of energy distribution alignment.

3: The paper Introduces the concept of free energy from statistical physics into the domain distribution alignment of DIL offers a unique perspective, and the design of the energy boundary loss and midline loss has good physical intuition. This method does not rely on historical data replay, greatly reducing storage pressure and avoiding potential data privacy risks.

Weaknesses:

 1: The method introduces six hyperparameters that require domain-specific tuning. Fig. 5 heatmaps show performance collapses under slight miscalibration (e.g., $\alpha$=0.1 or $\beta$=0.1), threatening real-world deployability. No automatic tuning strategy or sensitivity analysis across datasets is provided.

2: The "unknown domains" evaluation mixes true distribution shifts (e.g., CORe50 outdoor) with held-out domains from the same dataset (DomainNet), conflating generalization with memorization. Direct comparison to non-prompt methods on unknown domains is sparse, and the buffer-free claim is weakened by using a 10% validation set from the first domain, a form of implicit memorization.

---

> ### Author Rebuttal · Authors · 2026-03-29
>
> Thank you very much for your time and effort in reviewing our paper. We sincerely appreciate your feedback.
>
> ---
>
> **Q1.** The method depends on several tuned hyperparameters, and the heatmaps suggest that small miscalibration can hurt robustness, especially on unknown domains.
>
> **A1.** We agree that the hyperparameter design should be clarified. Importantly, these parameters are not equally sensitive. In our method, $\Theta$ and $\Delta$ are domain-adaptive statistics, $\alpha$ and $\beta$ are stable empirical coefficients from cross-dataset experiments, and $K$ is a low-sensitivity default parameter.
>
> For domain $i$, after training the domain-specific model $M_{pre}$, $P^i$,the target-class similarity $H^i(x)[y]$ exhibits a stable distribution. Let $S_{\max}^i$ be the maximum achievable in-domain target similarity. Then the theoretical lower bound of the energy is:
>
> $$LowerBound_i=-\ln(e^{S_{\max}^i})=-S_{\max}^i$$
>
> Since the role of $\mathcal{L}_{midline}$ is to make in-domain energies converge smoothly, the natural choice of the midline is the expected in-domain energy:
>
> $$\Delta_i=E_{x\sim\mathcal{D}^i}[E^i(x)] \approx -\ln\!\left(e^{\mu_{pos}^i}+(U-1)e^{\mu_{neg}^i}\right)$$
>
> Thus, $\Delta_i$ is not manually set, but estimated from the in-domain energy distribution, and it satisfies:
>
> $$-S_{\max}^i<\Delta_i$$
>
> To cover most in-domain samples, we define the upper tolerance threshold by the $3\sigma$ rule:
>
> $$\Theta_i=\Delta_i+3\sigma_E^i$$
>
> so that:
>
> $$\Delta_i<\Theta_i$$
>
> For out-of-domain inputs, similarities degenerate toward a shared mean $\mu^i$, giving the expected cross-domain energy:
>
> $$Upper\_Bound_i\approx-\ln\!\left(\sum_{y=1}^{U}e^{\mu^i}\right)=-\mu^i-\ln(U)$$
>
> Therefore, a sufficient separation condition is:
>
> $$-S_{\max}^i<\Delta_i<\Theta_i\ll-\mu^i-\ln(U)$$
>
> This shows that $\Theta_i$ and $\Delta_i$ are adaptive quantities derived from domain statistics rather than brittle tuned constants.
>
> For the hybrid score:
>
> $$F(x)=\frac{1}{\alpha}EF(x)+\frac{1}{\beta}DF(x)$$
>
> Proposition 1 shows that the two errors are approximately orthogonal, i.e., $Cov(\mathcal{E}_{EF},\mathcal{E}_{DF})\approx0$. Hence:
>
> $$Var(F)=\left(\frac{1}{\alpha}\right)^2Var(EF)+\left(\frac{1}{\beta}\right)^2Var(DF)$$
>
> Under minimum-variance fusion, the optimal ratio satisfies:
>
> $$\frac{\alpha}{\beta}=\sqrt{\frac{Var(EF)}{Var(DF)}}$$
>
> Empirically, cross-dataset experiments consistently place the optimum in a narrow range: $\alpha\in[0.7,0.9]$ and $\beta\in[0.4,0.6]$. Thus, $\alpha$ and $\beta$ are robust empirical coefficients, not domain-specific fine-tuned parameters.
>
> Finally, after $L_2$ normalization, CLIP features lie on the unit hypersphere, and:
>
> $$D^i(x)=\min_{j=1}^{K}\left(1-\cos(Z_v(x),M_j^i)\right)$$
>
> Hence, $K$ only controls prototype discretization granularity. Once the major semantic modes are covered, increasing $K$ brings only marginal gain, which matches our experiments.
>
> In summary,  $\Theta$ and $\Delta$ are automatically estimated, $\alpha$ and $\beta$ are stable across datasets, and $K$ can be set to a default value.
>
> 2.Concerns Regarding the Heatmap. We have thoroughly tested the hyperparameters across three tasks. Through these extensive experiments, we identified the empirical value intervals for the two parameters to be $\alpha \in [0.7, 0.9]$ and $\beta \in [0.4, 0.6]$. To further test the extreme values of these parameters, we did observe the performance drop around 0.1 that you mentioned. However, it is important to note that the performance only decreases by 2 to 3 percentage points, rather than exhibiting a sharp or catastrophic decline. Therefore, this method is robust and will not negatively impact practical deployment.
>
> **Q2:** "Unknown domain" conflation and the buffer-free claim.
>
> **A2.** Our task is defined as using the untrained data domain as the unknown domain. To clarify the experimental setup: CORe50 contains 11 domains in total. We use the first 8 indoor domains for incremental training and the remaining 3 outdoor domains as unknown domains. DomainNet contains 6 domains, where the first 5 domains (Clipart, Infograph, Painting, Quickdraw, and Real) are used as incremental training domains, and the last domain (Sketch) is treated as the unknown domain.
>
> Regarding the second point, on CDDB we reserved a small subset (10%) of the first domain only for hyperparameter selection at the beginning of training. This subset is not used as a replay buffer during subsequent domain-incremental learning. Throughout the actual training process, we strictly maintain a buffer-free setting, which can also be verified from our released code.
>
> **Q3.** Could the authors clarify if Spherical K-means was used?
>
> **A3:**  Thank you for pointing that out. In the current implementation, we do use the spherical K-means clustering algorithm; the consideration for using spherical K-means was to improve computational efficiency during inference. This will be explicitly stated in future revisions.

---

> > ### Author Rebuttal · Reviewer_BfkD · 2026-04-04
> >
> > We thank the reviewer for the thoughtful follow-up and clarification, which helped resolve our concerns.

---

> > > ### Author Response · Authors · 2026-04-04
> > >
> > > We are deeply grateful to BfkD for the thoughtful and constructive comments. These suggestions have provided us with clear and valuable guidance for further improving the paper.

---

### Official Review · Reviewer_3iDn · 2026-03-16

**Soundness:** 3
**Presentation:** 3
**Significance:** 3
**Originality:** 3
**Overall Recommendation:** 4
**Confidence:** 4

**Summary:**

This paper proposes HEDP (Hybrid Energy-Distance Prompt), a prompt-based framework for Domain Incremental Learning (DIL) that tackles two core challenges simultaneously: (1) mitigating inter-domain feature overlap during training, and (2) enabling reliable domain selection at inference for both seen and unseen domains. Built upon a frozen CLIP backbone, HEDP introduces two components: Energy Regularization Loss that composes of a boundary loss and a midline loss based on the Helmholtz free energy. At test time, per-domain energy factors and distance factors are fused into a unified hybrid factor that produces softmax weights over domain-specific prediction probabilities for the final classification. Experiments show improvements.

**Compliance With Llm Reviewing Policy:**

Affirmed.

**Final Justification:**

The rebuttal addresses my concerns, therefore maintaining my positive score.

**Key Questions For Authors:**

The setting is similar to domain generalization (DG), where source model trained on multiple source domains is expected to generalize to unknown distributions. The authors are encouraged to discuss the proposed method’s applicability to the DG setting, where weighted ensemble of different models [1] or domain predictors [2] have been proved effective.

[1] Wortsman M, Ilharco G, Kim J W, et al. Robust fine-tuning of zero-shot models[C]//Proceedings of the IEEE/CVF conference on computer vision and pattern recognition. 2022: 7959-7971.

[2] Li X, Min Y, Chen H, et al. Generalizing Vision-Language Models with Dedicated Prompt Guidance[J]. arXiv preprint arXiv:2512.02421, 2025.

**Limitations:**

yes

**Strengths And Weaknesses:**

Strengths:
- The idea of adopting Helmholtz free energy to model domain-level compatibility and regularize training is valid and inspiring.
- The proposed method is validated with both comprehensive ablation study and theoretical analysis.
- The paper is well presented and illustrated.

Weaknesses:
- The method needs to train independently on each domain, which requires clear domain labels and boundaries. Such assumption is very strong in real-world continual learning scenarios and may not hold. The authors are suggested to propose alternatives for scenarios without clearly divided sub-domains.
- The method involves many hyperparameters to decide (boundary, midline, etc.), and Fig 6 shows that different datasets favor different hyperparameter sets. While the paper uses a validation set to determine optimal hyperparameters, such hyperparameter sets may not still be optimal for the following unknown data distributions.
- A computational complexity analysis is needed to understand how the method scales with more domains.

---

> ### Author Rebuttal · Authors · 2026-03-29
>
> Thank you very much for your time and effort in reviewing our paper. We sincerely appreciate your feedback. Below, we respectfully provide our detailed responses to address your concerns.
>
> ---
>
> **Q1.** How does unknown-domain DIL differ from DG, and why is HEDP not simply a DG method? The concern is whether the proposed setting can be reduced to a static DG problem.
>
> **A1.** We sincerely thank you for raising this point. After carefully reading the two cited works, we found that WiSE-FT [1] performs ensembling in the parameter space between the zero-shot model and the fine-tuned model, whereas GuiDG [2] adopts an explicit domain classifier and uses Cross-Modal Attention to compute the weights of different experts.
>
> After a careful comparison, we note that HEDP can be understood as integrating the fine-tuned model from each domain. For the domain-classification component, instead of using a standard explicit domain classifier, we differentiate domain experts through an energy-based mechanism, and further weight each domain expert using both energy and distance factors. Moreover, optimizing the domain classifier with the energy regularization loss improves the orthogonality among domain experts, which is beneficial for subsequent expert-ensemble inference.
>
> We also find the ideas in these two works highly inspiring. In the revision, we will discuss them in the Related Work section and further explore how such ideas can be adapted to our setting.
>
> **Q2.** HEDP relies on explicit domain labels and clear domain boundaries. The concern is that such assumptions may not hold in realistic continual learning scenarios with ambiguous or evolving domains.
>
> **A2.** We appreciate this insightful comment. In practice, domain boundaries are often not clearly defined. As shown by our t-SNE visualization of samples from different domains, there exist confusing points and overlapping regions across domains. This is exactly why a mechanism beyond explicit boundary assumptions is needed.
>
> The advantage of our design is that the Helmholtz free energy function compresses complex feature patterns into a comparable scalar energy, where similar samples generally fall into lower-energy regions. Our motivation for introducing the energy regularization loss is precisely to better distinguish ambiguous scenario boundaries and improve domain discrimination under such overlap. We have discussed this issue in Section 5.3 of the paper.
>
> We agree that this is an important point, and we will make the discussion and limitation more explicit in the revised manuscript.
>
> **Q3.** HEDP depends on several dataset-sensitive hyperparameters. The concern is that validation-tuned settings may not transfer well to future unseen distributions.
>
> **A3.** Regarding these hyperparameters, $\Theta$ and $\Delta$ are determined by the specific feature characteristics of each domain, whereas $\alpha$ and $\beta$ represent empirical intervals established through extensive cross-dataset experiments. Cross-dataset experiments also demonstrate that the parameter $K$ can simply be set to a conventional default value. We will add the following detailed explanations to the Appendix of the revised manuscript.
>
> Our empirical conclusions, derived through mathematical analysis, can be found in reviewer BfkD's Q1 for a detailed derivation.
>
> **Q4.** The computational complexity of HEDP has not been clearly specified.
>
> **A4. Training Phase.** Learning a new domain requires no rehearsal of past data or previous parameters. Thus, the time complexity for training a single new domain is $\mathcal{O}(1)$ with respect to the number of domains $S$, ensuring excellent scalability.
>
> **Inference Phase.**
>
> - **Step 1 (Distance Factor):** CLIP feature extraction ($\mathcal{O}(C_{vit})$) and cosine distance to $K$ cluster centers across $S$ domains ($\mathcal{O}(K \cdot S \cdot d)$). Total: $\mathcal{O}(C_{vit} + K \cdot S \cdot d)$.
>
> - **Step 2 (Energy Factor & Prediction):** Forward passes through $S$ prompted models to obtain predictions and EFs. Complexity: $\mathcal{O}(S \cdot C_{vit})$.
>
> - **Step 3 (Hybrid Weight Fusion):** Calculating soft weights and aggregating predictions across $S$ domains. Complexity: $\mathcal{O}(S \cdot U)$, where $U$ is the number of categories.
>
>   **Total Inference:** Summing these steps and omitting lower-order terms, the asymptotic time complexity is dominated by the Transformer passes, resulting in $\mathcal{O}(S \cdot C_{vit})$, which scales linearly as $\mathcal{O}(S)$.
>
>
> ---
>
> At last, we sincerely appreciate your valuable feedback, and we will carefully consider all your suggestions to further improve our paper. We would be deeply grateful if you could kindly reconsider raising the score. Thank you very much!

---

> > ### Author Rebuttal · Reviewer_3iDn · 2026-04-03
> >
> > Thanks for the authors' rebuttal. I choose to maintain my positive score.

---

> > > ### Author Response · Authors · 2026-04-04
> > >
> > > We sincerely thank 3iDn for the positive feedback and for the encouraging recognition of the value of our rebuttal. Such affirmation means a great deal to us and gives us strong confidence in the direction and significance of our work. We are truly grateful for the 3iDn’s support and generous encouragement throughout this process.

---

### Decision · Program_Chairs · 2026-04-30

**Decision:**

Accept (regular)

**Comment:**

This paper proposes HEDP, a hybrid energy-distance prompt-based framework for domain incremental learning. The method introduces an energy regularization objective to improve domain separability and a hybrid energy-distance weighting strategy for inference-time domain selection. Experiments on CORe50 demonstrate effectiveness in mitigating catastrophic forgetting and improving generalization.

During the review process, several key concerns were raised, mainly regarding the reliance on clear domain boundaries, applicability beyond domain incremental settings, computational complexity, hyperparameter sensitivity, evaluation of unknown domains, and clarification of the clustering strategy and energy formulation. After the rebuttal, all reviewers acknowledged that their concerns had been fully resolved.

Overall, this work meets the requirements of the conference, but the authors should revise the manuscript to address the remaining issue, especially the clarification of attribute interpretation. For these reasons, I recommend acceptance.